# Chondrocyte Culture Parameters for Matrix-Assisted Autologous Chondrocyte Implantation Affect Catabolism and Inflammation in a Rabbit Model

**DOI:** 10.3390/ijms20071545

**Published:** 2019-03-27

**Authors:** Martin Sauerschnig, Markus T. Berninger, Theresa Kaltenhauser, Michael Plecko, Gabriele Wexel, Martin Schönfelder, Valerie Wienerroither, Andreas B. Imhoff, Philip B. Schöttle, Elizabeth Rosado Balmayor, Gian M. Salzmann

**Affiliations:** 1Department of Orthopedic Sports Medicine, Klinikum rechts der Isar, Technical University of Munich, Munich 81675, Germany; markus.berninger@gmx.at (M.T.B.); theresa@suetfels.com (T.K.); the.rabbithunter@gmx.de (G.W.); a.imhoff@sportortho.de (A.B.I.); p_schoettle@web.de (P.B.S.); gian.salzmann@kws.ch (G.M.S.); 2Trauma Hospital Graz, Unfallkrankenhaus der Allgemeinen Unfallversicherungsanstalt (AUVA), Teaching Hospital Medical University Graz, Graz 8010, Austria; michael.plecko@auva.at (M.P.); valerie.wienerroither@medunigraz.at (V.W.); 3Department of Trauma Surgery, Trauma Center (BGU) Murnau, Murnau 82418, Germany; 4Exercise Biology, Technical University of Munich, 80809 Munich, Germany; martin.schoenfelder@tum.de; 5Department of General Surgery, Medical University of Graz, Graz 8036, Austria; 6Experimental Trauma Surgery, Klinikum rechts der Isar, Technical University of Munich, Munich 81675, Germany; elizabeth.rosado-balmayor@tum.de; 7Gelenkzentrum Rhein-Main, Wiesbaden 65183, Germany; 8Musculoskeletal Centre, Schulthess Klinik Zurich, Zurich 8008, Switzerland

**Keywords:** matrix-assisted autologous chondrocyte implantation, cell passage, cell density, membrane-holding time, cytokines, matrix metalloproteinases, animal model

## Abstract

Cartilage defects represent an increasing pathology among active individuals that affects the ability to contribute to sports and daily life. Cell therapy, such as autologous chondrocyte implantation (ACI), is a widespread option to treat larger cartilage defects still lacking standardization of in vitro cell culture parameters. We hypothesize that mRNA expression of cytokines and proteases before and after ACI is influenced by in vitro parameters: cell-passage, cell-density and membrane-holding time. Knee joint articular chondrocytes, harvested from rabbits (*n* = 60), were cultured/processed under varying conditions: after three different cell-passages (P1, P3, and P5), cells were seeded on 3D collagen matrices (approximately 25 mm^3^) at three different densities (2 × 10^5^/matrix, 1 × 10^6^/matrix, and 3 × 10^6^/matrix) combined with two different membrane-holding times (5 h and two weeks) prior autologous transplantation. Those combinations resulted in 18 different in vivo experimental groups. Two defects/knee/animal were created in the trochlear groove (defect dimension: ∅ 4 mm × 2 mm). Four identical cell-seeded matrices (CSM) were assembled and grouped in two pairs: One pair giving pre-operative in vitro data (CSM-i), the other pair was implanted in vivo and harvested 12 weeks post-implantation (CSM-e). CSMs were analyzed for TNF-α, IL-1β, MMP-1, and MMP-3 via qPCR. CSM-i showed higher expression of IL-1β, MMP-1, and MMP-3 compared to CSM-e. TNF-α expression was higher in CSM-e. Linearity between CSM-i and CSM-e values was found, except for TNF-α. IL-1β expression was higher in CSM-i at higher passage and longer membrane-holding time. IL-1β expression decreased with prolonged membrane-holding time in CSM-e. For TNF-α, the reverse was true. Lower cell-passages and lower membrane-holding time resulted in stronger TNF-α expression. Prolonged membrane-holding time resulted in increased MMP levels among CSM-i and CSM-e. Cellular density was of no significant effect. We demonstrated cytokine and MMP expression levels to be directly influenced by in vitro culture settings in ACI. Linearity of expression-patterns between CSM-i and CSM-e may predict ACI regeneration outcome in vivo. Cytokine/protease interaction within the regenerate tissue could be guided via adjusting in vitro culture parameters, of which membrane-holding time resulted the most relevant one.

## 1. Introduction

As a cell-based method for the treatment of full thickness cartilage defects, autologous chondrocyte implantation (ACI)—first described in 1994—promised the dawn of a new era [1]. The first rush of enthusiasm was extenuated for complications such as graft hypertrophy and calcification soon to be redeemed by another hype coming up with the introduction of artificial cell carriers and novel culture methods [2,3,4]. Unfortunately, the problems solved by the alternation of ACI generations happened to be replaced by others. Awareness occurred to potential limitations within the boundaries of in vitro culture such as rapid phenotypic changes of passaged chondrocytes [5], indicating in vitro settings to claim major influence on cell differentiation. Cartilage defects are known to represent an assignable cause of osteoarthritis (OA)—the most common musculoskeletal disease—which entails pain and immobilization due to progressive cartilage deterioration imposed to the affected joint [6,7]. An imbalance of expression and activity of cytokines and matrix metalloproteinases (MMPs) is documented to play a pivotal role in OA pathophysiology [8]. However, basic mechanisms of production and release as well as principles of multiple interactions among such biochemical mediators within initial and progressive OA remain rather unclear [9,10,11]. 

Most recently, Boehme and colleagues presented a review of the literature focusing on onset and progression of human OA—with the much appreciated focus on the often-discussed growth factors as well as inflammatory cytokines [12]. This review discloses only fibroblast growth factor 2 (FGF2) to be capable of inducing all major key events of early OA, i.e., the proliferation of cartilage-resident cells, the degradation of extracellular matrix components and inflammation throughout known literature so far. This conclusion happens to be a major trigger for the further investigation of the basics of the pathophysiology of OA within the study presented here.

Halbwirth et al. reported that chondrocytes themselves play a major role in the pathophysiology of OA. They can produce a wide array of different MMPs and cytokines in dependence of their extracellular environment [13]. As the superfamily of zinc-dependent proteinases might take part in degrading as well as cell-redeeming processes, proinflammatory cytokines interact with them and with each other, thus directly as well as indirectly provide their share to tissue homeostasis, degrading, remodeling and repair, i.e., regeneration. Consequently, we hypothesize that inflammation and catabolism may play major roles when transplanting chondrocytes under clinical circumstances. Moreover, we conjecture that the in vitro culture settings employed during the preparation of the cells for ACI are of crucial importance for the in vivo outcome. 

Product preparation for ACI still lacks standardized in vitro parameters. The only standardization available is provided by the European Medicines Evaluation Agency’s (EMEA) guidelines defining quality, safety and efficacy. Consequently, ACI-developing companies often employ different parameters such as cell passage or membrane-holding time after seeding without any normative regulation. Thus, strong inter-company heterogeneity has been reported. However, in certain countries, cellular quantity has been defined to be not less than one million chondrocytes per 1 cm^2^ cartilage defect [14]. Strong evidence for why to use this specific number of cells is lacking. Moreover, different implantation localizations may need different cell densities, as cartilage thickness and cell density vary across the knee joint [15]. Furthermore, it is depicted that intraoperative handling of ACI products has a major effect on cell viability and therefore cell count [16]. In a similar model, it is shown that changes in such basic cell culture parameters largely impact chondrocyte differentiation and de-differentiation [17]. However, little is known about the complex cascades of pro-inflammatory cytokines and MMPs within ACI implants or their interactions with the joint’s environment [8,18,19]. The aim of the current analysis was to investigate mRNA expression patterns of pro-inflammatory cytokines (TNF-α and IL-1β) and matrix metalloproteinases (MMP-1 and MMP-3) before and after implantation of an ACI product in response to the variation of in vitro culture parameters (i.e., cell density, cell passage and membrane-holding time) in a rabbit cartilage defect model.

## 2. Results

### 2.1. Matrix Metalloproteinases (MMP-1 and MMP-3)

A proportional linearity between respective CSM-i and consecutive CSM-e values of proteinase expression occurred (Figure 1). 

Expression levels of MMP-1 and MMP-3 showed significantly higher results in CSM-i compared to CSM-e (*p* ≤ 0.001), as illustrated in Figure 2. Analyzing the effect of each in vitro culture parameter, it can be observed that increasing cell passage (P) generally resulted in lower MMP-1 and MMP-3 expression (Figure 3). Interestingly, this could not be observed for CSM-e with shorter membrane-holding time (T1, Figure 3). 

In this case, the expression of MMP-3 increased with increasing the cell passage. The membrane-holding time showed to have a crucial impact on the MMP-1 and MMP-3 expression. This effect was observed for both CSM-i and CSM-e. Longer membrane-holding time (i.e., T2 = 2 weeks) produced higher MMP-1 expression in CSM-i as well as in CSM-e. In the case of MMP-3, this was only true for CSM-i. For CSM-e, shorter membrane-holding time (i.e., T1 = 5 h) led to increased expression of MMP-3 if combined with higher cell passages (P3 and P5, Figure 3). Short membrane-holding time, i.e., T1 groups, combined with low cell densities (C1 and C2) and less frequent cell passaging produced a decreased MMPs expression (P3C1T1 and P3C2T1). All longer membrane-holding time (T2) combinations increased MMP-1 expression. On the other hand, short membrane-holding time combined with higher cell passage increased MMP-3 expression. Surprisingly, the cell density imposed a rather mild effect on the MMP expression. 

Overall, MMP-1 and MMP-3 showed the highest expression when lower cell passage was combined with longer membrane-holding time, i.e., P1 combined with T2. An overview of in vitro/in vivo (CSM-i NS CSM-e) proteinase expression patterns (MMP-1 AND MMP-3) showing the effect of cell passage (P) and membrane-holding time (T) is presented in Figure 3. 

### 2.2. Proinflammatory Cytokines (IL-1β and TNF-α)

For IL-1β, proportional linearity between respective CSM-i and consecutive CSM-e values occurred (Figure 4), while no such linearity could be shown for TNF-α. Expression levels of IL-1β were significantly higher among CSM-i when compared to their respective CSM-e (*p* ≤ 0.005).

The opposite was true for the expression of TNF-α (Figure 5). Only the combinations P1C2T1, P3C1T1 and P3C2T1 produced lower expression of TNF-α in CSM-e compared to CSM-i. 

An overview of in vitro/in vivo (CSM-i and CSM-e) cytokine expression patterns (IL-1β and TNF-α) divided for cell passage (P) and membrane-holding time (T) is given in Figure 6. Short membrane-holding time T1 groups combined with low cell density and cell passage produced significantly lower IL-1β expression in CSM-i when compared with long membrane-holding time T2 combinations. In contrast, among CSM-e T2 combinations, significantly lower IL-1β expression was found when compared to T1 groups. 

TNF-α showed significantly lower expression in T2 compared to T1 within CSM-i and CSM-e. Lower cell passage (P1 and P3) led to stronger TNF-α expression, while cell density appeared to be of mild effect.

### 2.3. Control

In the uncolonized membranes, no mRNA could be detected by analyses after mRNA isolation—neither after two weeks of in vitro cultivation under standard conditions, nor after the 12-week in vivo phase. In addition, mRNA quantification was below the detection limit in the empty chondral defects. 

## 3. Discussion

The data presented here demonstrate that in vitro and in vivo expression patterns of selected cytokines and matrix metalloproteinases within ACI regenerates are strongly influenced by varying in vitro laboratory parameters. The MMPs and IL-1β showed similar expression trends, while TNF-α patterns appeared to be different. Strong initial in vitro expression of the investigated parameters is generally reduced 12 weeks post-implantation with the exception of TNF–α, which was found to increase within tissue regenerates over time. With the exception of TNF-α, in vitro mRNA expression of the remaining investigated targets was linearly connected to the in vivo values. While increasing cell-passages did not strongly affect cytokine or MMP expression particularly in vitro, the parameter membrane-holding time clearly showed the strongest impact on the targets investigated in this study. Prolonged membrane-holding time clearly increased in vitro cytokine and matrix metalloproteinase expression. In vivo, however, longer membrane-holding time decreased these expression patterns. Chondrocyte quantity (cell density) per matrix did not significantly affect the selected outcome parameters in this study. The here-presented study explicitly investigated mRNA expression patterns of MMPs and cytokines and their potential dependency of in vitro cell culture settings. Within further studies, it would be very interesting to further investigate if these results are true for the activated protein level fraction of MMPs as well. The latter notion combined with the results presented here could provide further insights into the cascades of early healing response and/or tissue degradation following cartilage damage and potentially deterioration of joint homeostasis.

Cartilage lesions are diagnosed and treated with increasing frequency. Nowadays, it is considered that morphological repair of defective cartilage may be only one part of reconstitution of the joint—considered as an organ embedded in a harmonized system. Related to physiological homeostasis, cytokines, growth factors, and catabolic and anabolic proteins play other important roles in maintaining the function of the joint free of pain and full activity ability. A balanced integrity of i.a. hyaline cartilage, menisci, subchondral bone, ligament-stability, synovial fluid and synovial layer is essential for this sensitive environment. Currently, autologous chondrocyte implantation is considered the standard approach to cover large diameter full thickness cartilage defects among the knee joint. However, specific guidelines—which often arise from localized working groups [20]—recommend treatments with ACI already at defect areas of 2.5–3 cm^2^ with the idea to gain an optimal tissue regeneration to avoid early onset OA [21]. ACI is usually accomplished by performing a cartilage biopsy that is subsequently sent to a laboratory to undergo further processing. This processing involves cell culture including multiple passaging/population doublings. The in vitro parameters cell passage number, cell density and membrane-holding time still lack standardized values. In addition, ACI variants are encumbered by drawbacks such as graft hypertrophy, delamination, absent maturation/implementation or cellular phenotypic dedifferentiation [5]. We hypothesize that ACI graft maturation, differentiation, integration and final long-term function may be strongly dependent on the initial and further cell quality of the autologous chondrocytes. In addition, these factors may likely be also affected by the existing joint homeostasis where the graft is implanted. Finally, optimal graft production may affect the final transplant outcome with optimized joint homeostasis. In a previously published study, it could be shown that cell culture parameters are able to strongly impact transplant performance in vitro and in vivo [17]. Here, we could show that such change in culture conditions also affected pro-inflammatory cytokines and matrix metalloproteinases in a significant manner. Imperfect regenerated tissue may arise and/or may even harm existing joint homeostasis [5,22]. In that way, Filardo et al. found overall poor outcome following matrix-assisted autologous chondrocyte transplantation (MACT) in osteoarthritic knee joints in a five-year follow-up [23]. This could be seen as a consequence of cytokines and cartilage-degrading enzymes prevalent in OA knee joints, which potentially impose a negative influence on an ACI regenerate. However, already pre-arthritic chondrocytes taken during initial biopsy to perform ACI may be less chondrogenic, proliferative and resistant to stress such as inflammation when compared to cells obtained from joints of healthy subjects. Hence, these cells may be less able to produce stable hyaline cartilage, while ongoing intraarticular inflammation may constantly deteriorate their differentiation. Comparably, Albrecht et al. described that IL-1β expression negatively influenced clinical outcomes at 24 months (Brittberg score) and 60 months (Brittberg and VAS scores) after surgery (*p* < 0.05) in a five-year ACI outcome study [24]. 

It is currently known that chondrocytes are capable of producing cytokines, metalloproteinases and catabolic proteins that negatively impact the tissue via autocrine and paracrine pathways [25,26]. The ability to regulate expression strength of principal cytokines such as IL-1β and TNF-α and/or proteinases such as MMP- and MMP-3 within the cell–matrix construct may lead to an improvement of ACI performance. 

Efficacy of graft remodeling and integration requires a balanced activity of proteinases and their inhibitors while preponderance of cartilage-degrading proteins such as MMP-1/MMP-3 may cause deterioration of ACI constructs in vivo [27,28]. In our study, the expression of both MMPs was strongly upregulated by prolonged membrane-holding time in vitro while the remaining parameters had considerably less effect. These significantly higher expression levels of both MMPs in vitro indicated that the implanted chondrocytes have already begun with remodeling in their three-dimensional environment. After 12 weeks in vivo, these processes seemed to be already partially completed. In contrast, Vasara et al. reported increased MMP-3 expression in the synovial fluid of patients who have undergone ACI one year prior concluding that the increased values would indicate either a graft remodeling or early degeneration of the implanted tissue [18].

The parameter membrane-holding time has shown various effects in previous studies. Kawamura et al. revealed that the implant could better cope with the mechanical stress in vivo by an early three-dimensional pre-cultivation of the chondrocytes on a membrane [29]. A longer in vitro culture time (i.e., four weeks) in the study of Lee and colleagues was beneficial concerning histomorphometry and certain mechanical properties. In the study, the authors described improved defect filling with hyaline-like cartilage for long (four weeks) compared to short culture period (12 h) [30]. In contrast, Gavenis et al. showed no improvement by a prolonged membrane-holding time [31]. Today, it is known that a short membrane-holding time has beneficial effects on graft maturation and differentiation [17]. This results in matrix deposition surrounding the cells, which allows the transmission of biomechanical signals through the matrix rather than onto the naked cell.

The influence of cell density on ACI performance in vivo has been controversially discussed. Concaro et al. found the quality of human chondrocytes within tissue-engineered regenerates to improve with increasing cell density [32]. Willers et al. reported no significant differences after comparing various cell densities in m-ACI [33]. In a recent study using an osteochondral rabbit animal model, no positive effect on cartilage repair is shown in vitro and in vivo between different cell seeding densities [34]. The data presented here show cell density to impose a rather mild effect on the investigated expression patterns, which offers the possibility to normalize the broad claim of complex cell culturing methods towards reduced overall cell yield. Reduced yield necessitates less population doublings and reduced in vitro time, which pictures a worthwhile improvement to the vast majority of cartilage repair techniques [2].

The expression of IL-1β was strongly upregulated in CSM-i and downregulated in CSM-e by prolonged membrane-holding time. Again, cell passage and cell densities showed less effect. TNF-α expression was reduced following prolonged time on the membrane. Furthermore, this effect was enhanced with increasing passage. 

Higher passage is known to have highly deteriorating effects on chondrocyte phenotypes. Kang et al. described passage 5 to exhibit a strong expression of type I collagen [35]. The effect of in vitro pre-cultivation of chondrocytes on in vivo inflammatory processes was previously examined histologically and immunohistochemically by Luo et al. [36]. They demonstrated that inflammatory reactions post-implantation could be alleviated by prolonged in vitro pre-cultivation of chondrocytes. The reasons for early activation of inflammatory cascades might be a migration of blood cells into the membrane as well as interactions with components of the matrix. Both factors could possibly be limited by an early and adequate protective matrix production through a prolonged membrane-holding time.

Noting paramount differences between in vitro and in vivo data, as well as the exception of TNF- α, a trend for reduced catabolism and inflammation occurs when cells are less passaged and foremost cultured for a short period of time within a three-dimensional matrix prior to re-implantation. Regarding such transplant information, it can be considered that ACI products that are in current clinical use may have varying in vivo performance regarding their differentiation potential and cytokine profile, which are both closely connected to the previous in vitro cultivation protocols [14].

## 4. Materials and Methods 

This study was carried out in accordance with the institutional guidelines. In addition, the study was carried out in compliance with national and international laws (Directive 86/609/EEC, German animal welfare law, FELASA guidelines) and the protocol was approved by the district government of Bavaria (District Government of Upper Bavaria, Munich, Germany, File Number: 55.2-1-54-2531-65-07). Within the procedures involving animal care and treatment, all efforts were made to minimize suffering. The here-presented research draws from the study protocol published in the work of Salzmann et al. [17], but seeks to investigate a completely new hypothesis.

### 4.1. Study Design

To investigate expression patterns of pro-inflammatory cytokines and matrix metalloproteinases in matrix-assisted ACI prior to and after a 12-week in vivo period, articular chondrocytes of the right knee joints of female post-puberty New Zealand White rabbits (*n* = 54 + *n* = 6 control) were harvested from the trochlea and cultured using a varying set of in vitro parameters. Chondrocytes were sub-cultured up to passage 1 (P1), 3 (P3) or 5 (P5). Subsequently, chondrocytes were seeded on the 3D collagen membrane (Chondro-Gide^®^, Geistlich, Wollhusen, Switzerland) with a volume of approximately 25 mm^3^ at different cell densities (i.e., 2 × 10^5^/matrix = C1, 1 × 10^6^/matrix = C2 or 3 × 10^6^/matrix = C3). Next, the cell-seeded 3-D collagen membranes were cultured for two different time periods (i.e., 5 h = T1 and 2 weeks = T2) to again simulate differing in vitro membrane-holding time periods. Consequently, the setup presented here is promising in covering some of the multiple in vitro factors utilized by the industry more or less lacking evidence-based standards, all of which are regarded to be complex interacting, while representing key player potential in the ongoing struggle to yield the best possible clinical results.

An alternating combination of the named in vitro parameters (i.e., passage, cell density, and membrane-holding time) resulted in a total of *n* = 18 different cell culture parameter-related groups. Figure 7 provides a rough overview of the study design and experimental groups resulting from all possible different combinations of used cell culture conditions. A showcase overview of orchestration of one exemplarily selected cell seeded matrix (CSM) is given in Figure 8. 

In preparation for autologous CSM implantation at the left knee joint, four identical membranes/study animal were assembled with regard to cell passage, seeding density and membrane-holding time. The 4 membranes were separated into 2 identical pairs at the time of in vivo implantation. One pair was used to gather pre-operative in vitro data (CSM-i—this membrane was directly taken out of culture and processed for further analysis). The other pair was implanted in the rabbits’ left knee joint. Twelve weeks post-implantation, the cell-seeded matrix was harvested and used for further analysis. This membrane corresponds to the in vivo regenerate post-implantation i.e., CSM-e. 

### 4.2. Animal Model, Cell Culture and Matrix Preparation

All procedures involving animal treatment were performed respecting national and international laws and policies (DIRECTIVE 86/609/EEC; German animal welfare law; FELASA guidelines). The protocol was approved by the local government of upper Bavaria (number 55.2-1-54-2531-65-07). Female New Zealand White rabbits (Crl: KBL(NZW)) of approximately 3.5 kg were obtained from Charles River (Kisslegg, Germany). All rabbits were postpubescent with closed growth plates, which was verified by applying the Stroh sign [37]. Rabbits were kept under standard housing conditions with controlled temperature and lighting cycles. The animals were housed in cages with water and food ad libitum. Surgeries and subsequent treatments were performed in congruence to the protocol described already [17]. In brief, after shaving and disinfection, the right knee joint was opened via medial para-patellar arthrotomy under sterile conditions and the patella was then displaced to the lateral side. Hereafter, full-thickness cartilage pieces were harvested from the non-weight-bearing part of the trochlea femoris and the joint was then closed in layers. Carprofen (4 mg/kg once a day) and buprenorphine (0.03 mg/kg twice a day) were subcutaneously administered as post-operative analgesics for 3 days and the animals were then further kept under standard housing conditions. All animals reached full weight-bearing within one day postoperative. 

Articular chondrocytes were isolated from the collected cartilage pieces following a previously described protocol [38]. In short, cartilage pieces were washed several times with PBS and subsequently sectioned in small pieces (approximately 1 mm^3^). Next, cartilage pieces were minced and digested by means of trypsin and collagenase enzymatic action. For tissue digestion, cartilage pieces were firstly incubated in 0.25% trypsin (Biochrom, Berlin, Germany) for 30 min followed by a final incubation in 2.5 mg/mL collagenase A (Roche, Mannheim, Germany) for a maximum of 6 h. The mixture was centrifuged and the resulting cell pellet resuspended in cell culture medium (Dulbecco’s modified Eagles medium (DMEM), supplemented with 10% fetal bovine serum, 1% penicillin/streptomycin and 1% glutamine; all reagents from Biochrom, Berlin, Germany). Isolated chondrocytes were seeded in 25 cm^2^ flask until approximately 80% confluency was reached. 

For the preparation of CSMs, chondrocytes in passages 1, 3 and 5 were used each at densities of 2 × 10^5^, 1 × 10^6^ and 3 × 10^6^ cells/matrix. To perform the cell seeding, the desired cell number was suspended in 20 µL of cell culture media and directly added to the cell-adhesive side of the 3D Chondro-Gide^®^ collagen membrane. Subsequently, cell-loaded membranes were further cultured for either 5 h (T1) or 2 weeks (T2) before in vivo implantation. Thus, two different in vitro membrane-holding times were generated. 

For the autologous CSMs in vivo implantation, a medial para-patellar arthrotomy was performed on the left knee. The procedure followed was congruent to the one already described above for cartilage harvesting. Two chondral defects (4 mm in diameter and 2 mm in depth) were created in the left knee of each animal within the center of the trochlea groove. The defects were rinsed with sterile saline and two identical CSMs were rapidly placed in each defect by press-fitting. Care was taken to place the cell-adhesive and porous side of the collagen membrane directly in contact with the bottom of the defect. Per animal, two different study groups (i.e., CSM from two different groups) were implanted. The defects were then sealed with fibrin glue (Baxter, Vienna, Austria), whereby a certain fixation of the membrane could be obtained. Next, the patella was relocated and the joint was bended and stretched in full range of motion ten times. The patella was dislocated again to revise the fixation. Finally, the joint was closed in layers. An original high-resolution intraoperative picture of our press-fit implanted membrane is given in Figure 9. Further details on the operative procedure and CSM implantation can be consulted within the antecessor publication [17]. For the six rabbits of the control group, two chondral defects were created in the trochlea of the left knee of each animal. In each case, one defect was left empty while the other one was filled with an uncolonized membrane. Animals were housed under standard conditions for further 12 weeks post-operative. Moreover, animals were allowed to apply full weight on the implanted area. 

### 4.3. RNA Extraction

CSM-i and CSM-e samples were harvested from in vitro culture or in vivo experiments at indicated observation times. Subsequently, samples were thoroughly washed with sterile PBS and used for RNA isolation. Total RNA was isolated using the NucleoSpin RNA II^®^ kit (Macherey-Nagel, Düren, Germany). Prior to RNA isolation, the CSM samples were mechanically homogenized and cells were lysed by incubation in RA1 lysis buffer containing guanidinium thiocyanate and β-mercaptoethanol. Subsequent RNA isolation was performed according to the NucleoSpin RNA II^®^ kit instructions protocol provided by the manufacturer for total RNA purification from cultured cells or tissue. RNA concentration and purity was assessed utilizing the NanoDrop ND-1000^®^ spectrophotometer (PEQLAB, Erlangen, Germany). RNA was reverse-transcribed within the qRT-PCR procedure utilizing a one-step detection system as described below.

### 4.4. Quantitative Polymerase Chain Reaction (qRT-PCR)

Expression profiles were assessed via one-step qRT-PCR. For this experiment, a Quanti-Tect^®^ SYBR^®^ Green RT-PCR kit (Qiagen, Hilden, Germany) was used following the manufacturer’s instructions. A Rotor-Gene-6000^®^ (Corbett Life Science, Qiagen, Hilden, Germany) was employed as detection system. Primers were designed using the Primer-3 software, as previously described [39], and tested with the Basic Local Alignment Search Tool (BLAST). The rabbit primer sequences were MMP-1, 5′- TCA GTT CGT CCT CAC TCC AG -3′ (forward), 5′- TTG GTC CAC CTG TCA TCT TC -3′ (reverse); MMP-3, 5′- TCC CTG GGT CTG TTT CAC TC -3′ (forward), 5′- CAC TGC TGA AGG AAG AGA TGG -3′ (reverse); IL-1β, 5′- TAC AAC AAG AGC TTC CGG CA -3′ (forward), 5′- GGC CAC AGG TAT CTT GTC GT -3′ (reverse); TNF-α, 5′- GGC TCA GAA TCA GAC CTC AG -3′ (forward), 5′- GCT CCA CAT TGC AGA GAA GA -3′ (reverse); HPRT-1, 5′-CTT TGC TGA CCT GCT GGA TT-3′ (forward), 5′-GCT TGA CCA AGG AAA GCA AG-3′ (reverse); and β-Actin, 5′-CAG CGG AAC CGC TCA TTG CCA ATG G-3′ (forward), 5′-TCA CCC ACA CTG TGC CCA TCT ACG A-3′ (reverse). The 10 μL reaction mix consisted of 1.0 μL mRNA template, 0.1 μL RT-Mix, 5.0 μL 2xQuanti-Tect^®^ SYBR^®^ Green RT-PCR Master Mix, 0.4 μL of each primer (forward, reverse) and 3.1 μL RNase-free water. Rotor-Gene^®^ settings involved a reverse-transcription step at 50 °C (30 min), activation of the polymerase reaction at 95 °C (15 min), followed by 38 cycles of denaturation at 94 °C (15 s), primer annealing at 60/62 °C according to primer specific requirements (30 s), elongation at 72 °C (30 s) and fluorescence detection at 80 °C (15 s). A board of β-Actin and HPRT-1 was used as internal control and reactions were performed in duplicate followed by a specific dissociation curve analysis of each assay at 60 °C to 95 °C (30 s). Quantification of mRNA expression was assessed by the 2^−ΔΔ*C*t^ method [40].

### 4.5. Statistical Analysis

Statistical analysis was performed using the software package SPSS (Version 20; SPSS Inc., Chicago, IL, USA). Skewed data distribution was normalized by applying natural logarithm transformation. Linear mixed regression models were utilized for the analysis of quantitative parameters. In this term, marginal means for the factor variables cell passage, cell density and membrane-holding time were calculated. In a further detailed analysis, post-hoc comparisons of factor-level combinations (3 × 3 × 2) were conducted, depending on previous (overall) significance testing. Group comparisons noted within the results section are all significant at a two-sided 0.05 level of significance if not described differently. 

If our results yielded statistical significance, we explicitly refer to this in the Results Section. If not mentioned explicitly, the results are not statistically significant. To retain a maximum of power within the multiple comparisons being conducted in this explorative study, no adjustment of alpha-error level was conducted. For the purpose of graphical visualization, only the parameters cell passage and membrane-holding time were applied since the mathematical impact of the parameter cell density was generally much less effective when compared to cell passage and membrane-holding time.

## 5. Conclusions

Catabolism and inflammation can be regarded as an emerging field of interest when operatively treating chondral and osteochondral lesions. This is particularly true when complex and cost-intensive in vitro culturing has to be performed to realize efficient ACI. Data analysis of this experimental animal study clearly displayed that significant differences for the expression of MMPs, IL-1β and TNF-α appeared when simple cell culture parameters were changed. With the exception of TNF-α, these were linearly connected when single gene expression was concerned, producing a potential tool to forecast in vivo performance. Such data offer further insights to hypotheses stating joint homeostasis to be strongly interacting with an evolving tissue regenerate. Specific expression patterns in turn can be potentially steered or at least “guided” via shifting in vitro settings.

The consolidated findings of this study can be regarded as one small further step directing the potential of optimizing the complex interactions of proteolytic enzymes and inflammatory cytokines to constitute ideal conditions for next generation ACI.

## Figures and Tables

**Figure 1 ijms-20-01545-f001:**
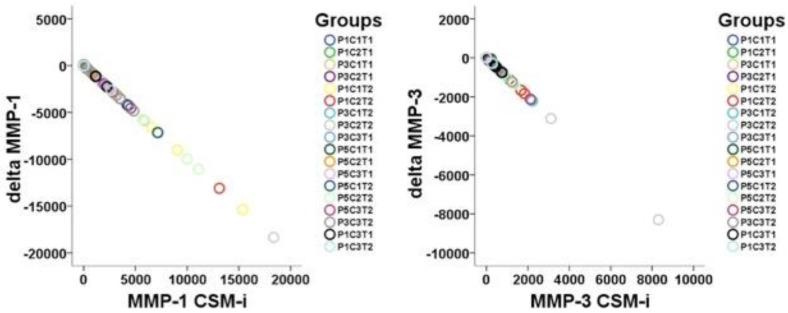
Visualization of linearity between CSM-i and the mathematical difference of CSM-e minus CSM-i (Δtarget) of all P, C, and T assemblies possible (*n* = 18) for MMP-1 and MMP-3.

**Figure 2 ijms-20-01545-f002:**
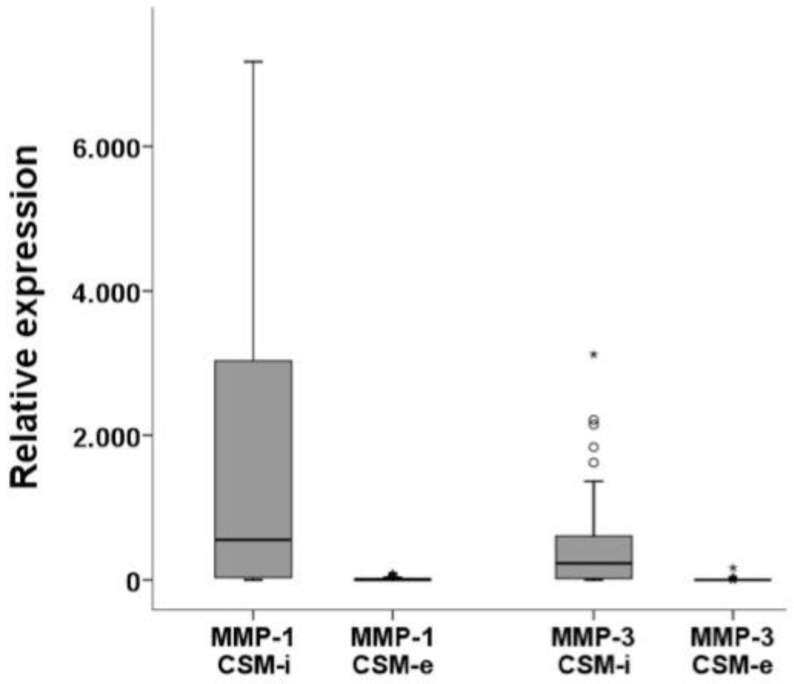
Relative expression levels of matrix metalloproteinases comparing CSM-i to CSM-e. Outliers are marked separately with *.

**Figure 3 ijms-20-01545-f003:**
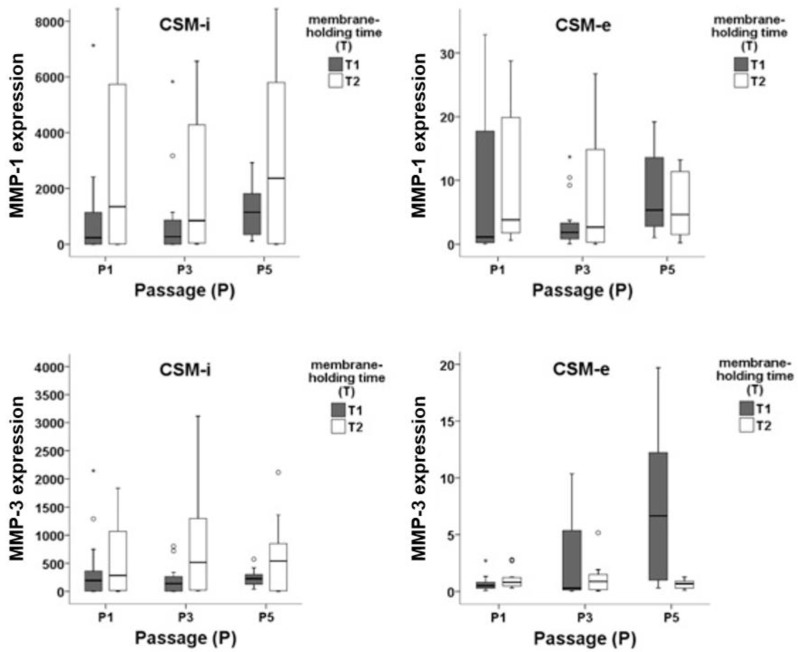
mRNA expression patterns of MMP-1 and MMP-3 within in vitro (CSM-i) and in vivo (CSM-e) membranes subdivided by cell passage 1 (P1), 3 (P3), and 5 (P5) and membrane-holding time 5 h (T1) and two weeks (T2). Outliers are marked separately with *.

**Figure 4 ijms-20-01545-f004:**
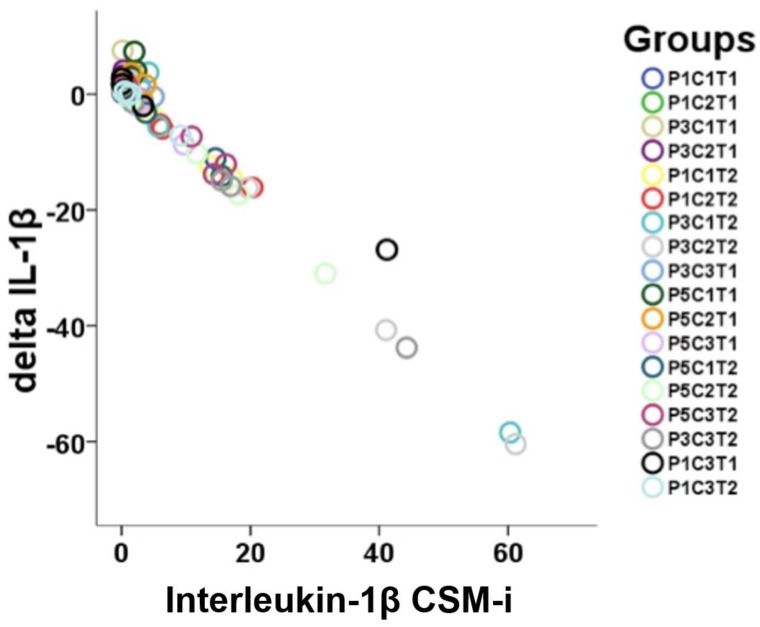
Visualization of linearity between CSM-i and the mathematical difference of CSM-e minus CSM-i (Δtarget) of all the P, C, and T assemblies possible (*n* = 18) for IL-1β.

**Figure 5 ijms-20-01545-f005:**
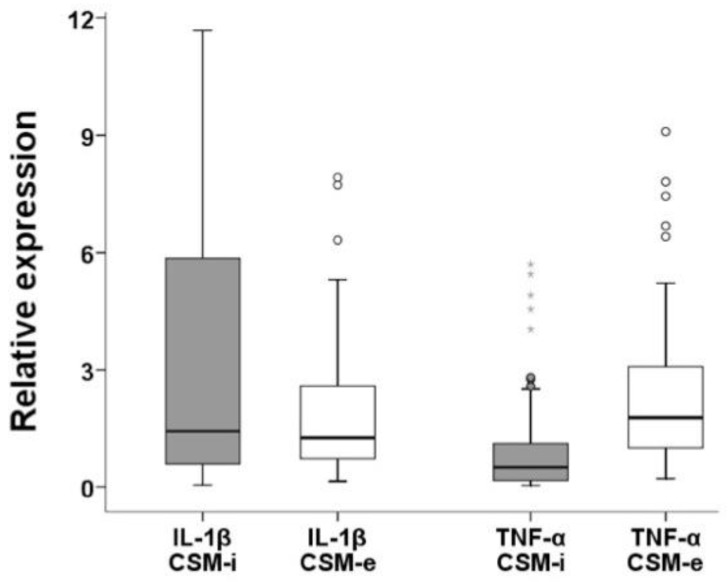
Relative expression levels of pro-inflammatory cytokines (IL-1β and TNF-α) comparing CSM-i to CSM-e. Outliers are marked separately with *.

**Figure 6 ijms-20-01545-f006:**
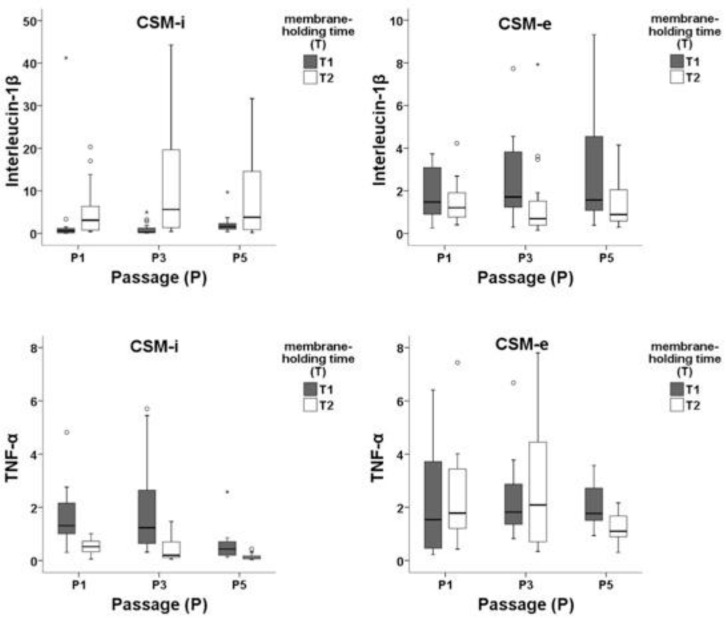
Expression patterns of IL-1β and TNF-α within in vitro (CMS-i) and in vivo (CMS-e) membranes subdivided by cell passage 1 (P1), 3 (P3), and 5 (P5) and membrane-holding time 5 h (T1) and two weeks (T2). Outliers are marked separately with *.

**Figure 7 ijms-20-01545-f007:**
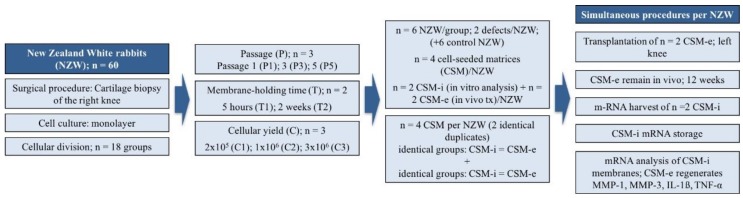
Study design overview. A total of 60 New Zealand White Rabbits (including 6 rabbits as control group) were included. From 54 included rabbits, harvested chondrocytes were divided into 18 different groups respecting variations of passages, cell count and membrane-holding time. Four cell-seeded membranes (CSM) per animal were conducted, two of which were identical according to passage (P), cell count (C) and membrane-holding time (T). Those duplicates were then split, two of which were re-transplanted simultaneously (ergo from different groups) and the other two were harvested from in vitro culture to isolate specific mRNA. Transplanted CSMs were harvested after a 12-week in vivo period and mRNA was isolated directly after sacrifice. Finally, qPCR was performed to analyze expression patterns of MMP-1, MMP-3, IL-1β and TNF-α.

**Figure 8 ijms-20-01545-f008:**
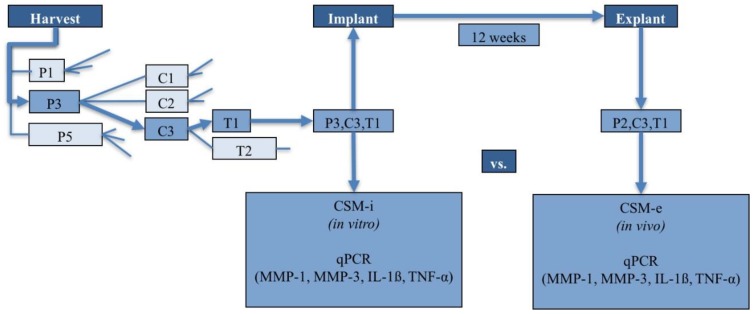
Example-orchestration of a CSM (showcase pathway with luminous arrows).

**Figure 9 ijms-20-01545-f009:**
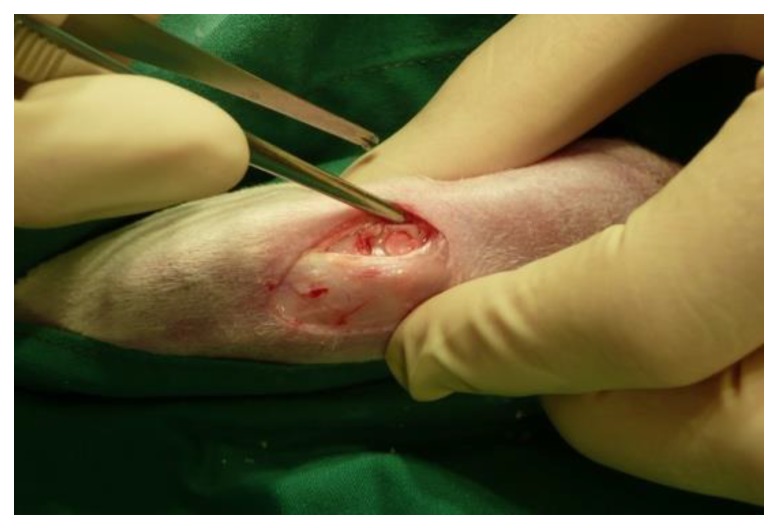
Intraoperative view on the chondral defect filled with the implanted membrane.

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
