# Peer review of "Chondrocyte Culture Parameters for Matrix-Assisted Autologous Chondrocyte Implantation Affect Catabolism and Inflammation in a Rabbit Model"

_ijms, 2019, doi:10.3390/ijms20071545_

Round 1

Reviewer 1 Report

Title:

Chondrocyte culture parameters for matrix-assisted chondrocyte implantation affect catabolism and  inflammation in a rabbit model

By Sauerschnig et al.,

The study seems obviously to be based on the in vivo experiments performed in Salzmann et al., published 2011 in „Biomaterials“. In addition to this already published study the gene expression of four genes (TNFα, IL-1β, MMP1, -3) was assessed and discussed as the experimental base for the present publication. The study would benefit from some histological images to visualize the defect healing. Hence, the study hampers from the fact that there is not relation between the cytokine/ MMP regulation and the overall healing outcome. The investigated cytokines and MMPs might play an important role in natural early healing or contribute substantially to the late remodelling process during haling.

The study would also benefit from a control group which is acking (empty defects). This group is important because even in the case when creating a critical defect size in the rabbit you will usually have complete intrinsic defect healing.

The selected passages (P1-P5) should be discussed in more detail: literature recommends passsage 4 as a threshold in regard to chondrocyte dedifferentiation, please refer in your manuscript to e.g. Kang SW et al., 2007 (PMID17851169) showing this exactly for rabbit-derived chondrocytes….

Was an autologous approach used in the study? It is not clearly enough stated in the manuscript. Otherwise it is recommended to write „allogenic“ or indeed „autologous rabbit model“ e.g. in the title. It is also not completely clear whether indeed a chondral defect model was used and not an osteochondral one. It should be stated in the abstract.

It is not always detailed whether the mentioned differences between data are indeed significant or not.

The terms „implantation/implanted“ and „transplant/ation“ (e.g. line 190 and line 192) seems to be mixed throughout the manuscript, please provide a rationale.

Abstract

„contribute“ write „to“ instead of „in“

Line 30: the dimensions (thickness, diameter: volume) of the collagen membrane should be stated to get a relation to the cell numbers listed here

Line 32: „Two defects“, please write already here whether they are „critically sized (dimension!)“ or not. Compare „4 mm“ line 345, how about the depth?

Line 33: Write also already here, where the defects are exactly localized (trochlea, med./lat. condyles?)

Introduction

TNFα, IL-1β, MMP1-3: please explain why exactly these targets were investigated in the study. Please consider also the important role of proinflammatory cytokines in the early healing process and that of MMPs in the later remodeling process. Proinflammatory cytokines are not exclusively „bad guys“

Fig. 3:

Y axis: if correct, write gene expression.

Fig. 7: membrane holding time n=2 – is it sufficient for statistics?

Discussion

Comment on the fact that gene expression is investigated in regard to MMPs but not activated protein levels.

First paragraph: The different regulation of TNFα in contrast to IL-1β is highly interesting and should be discussed in more detail (develop a hypothesis).

„membrane holding time“ why is this parameter so important: provide a hypothesis.

How about the effect of particular the biomaterial used in this study and the previous study from 2011? It should be discussed why the authors could generalize their results to other biomaterials?….

Between the first and second paragraph oft he discussion there seems to be a break – please create a bridge.

Line 230: The sentence requires a citation for support.

Line 236: „same osteochondral rabbit animal model“. Be aware that elswhere in the manuscript the used model is designated as „chondral“….please correct!

Materials and methods

4.1.: „femal post puberty“. Please mention the approximate age of the animals and whether epiphyses are closed or not as a parameter to estimate roughly musculoskeletal maturity. Compare line 317: 3.5 kg BW.

Line 275: provide exact dimension of the membrane it required to understand the cell numbers, compare line 337

Why was no autologous rabbit serum used for culturing? Compare line 334: fetal bovine serum.

Line 323: „full thickness cartilage pieces“. How much cartilage was explanted for cell isolation? Was it only removed from a non-weight bearing area of the trochlea (e.g. uppermost cranial part?). Or was the whole joint cartilage removed? How about the welfare of the animals after removing completely the trochlear joint cartilage?

Line 341: membrane-holding time. Was the seeding efficacy determined. How many cells survived and adhered on the scaaffolds?

Line 345: add „diameter“ and describe the depth of the defect.

Line 345: „chondral defects“ I am not sure whether the effects are indeed chondral?

Line 347: „press fitting“ – I think it is not possible to achieve in a chondral defect in the rabbit model sufficient fixation by press fitting and also not by fibrin sealant…it would be possible in an osteochondral defect in the rabbit.

Please show an OP image of the defect with the scaffold.

Line 349: please discuss the effect oft he fibrin because it is probably human-derived.

Author Response

Point-by-point response

Reviewer#1

The study seems obviously to be based on the in vivo experiments performed in Salzmann et al., published 2011 in „Biomaterials“. In addition to this already published study the gene expression of four genes (TNFα, IL-1β, MMP1, -3) was assessed and discussed as the experimental base for the present publication. The study would benefit from some histological images to visualize the defect healing.

Thank you for your remark. Although we definitely agree with the reviewer that additional histological images would have been interesting, we had a narrower focus on the expression of cytokines and proteases in this part of our project. As the overall study specimen were euthanized already, we cannot provide histological images at this point, due to ethical reasons. Subsequent studies in this line of research could however include histology and we would like to respect your remark for future work.

Hence, the study hampers from the fact that there is not relation between the cytokine/ MMP regulation and the overall healing outcome. The investigated cytokines and MMPs might play an important role in natural early healing or contribute substantially to the late remodelling process during haling. 

Yet, aim of this study was to analyse early cytokine expression in relation to different culture parameters of chondrocytes. Healing outcome was not the aim. Thus, we did not check for that. The whole study including ethical approval, statistical analysis, grant, students etc. was designed for that. We totally agree with the reviewer that healing outcome would be interesting - also to us. Yet, we would have to re-do the whole study again. However, we would like to extend our future research on this specific topic.

The study would also benefit from a control group which is acking (empty defects). This group is important because even in the case when creating a critical defect size in the rabbit you will usually have complete intrinsic defect healing.

The authors indeed agree with the reviewer and added the results of our control group that was already mentioned in 4.1 Study design, line 291, as well as the description of the intraoperative preparation to the associated sections (line 171-175 and 389-391):

2.3 Control

In the uncolonized membranes, no mRNA could be detected by analyses after mRNA isolation - neither after 2 weeks of in vitro cultivation under standard conditions, nor after the 12-week in vivo phase. Also, mRNA quantification was below the detection limit in the empty chondral defects.

4.2 Animal model, cell culture and matrix preparation

For the six rabbits of the control group, two chondral defects were created in the trochlea of the left knee of each animal. In each case one defect was left empty while the other one was filled with an uncolonized membrane.

We also adapted the legend of figure 7 (line 319-322):

A total of 60 New Zealand White Rabbits (including 6 rabbits as control group) were included and from 54 rabbits were included and harvested chondrocytes were divided forming 18 different groups respecting variations of passages, cell count and membrane-holding time.

The selected passages (P1-P5) should be discussed in more detail: literature recommends passsage 4 as a threshold in regard to chondrocyte dedifferentiation, please refer in your manuscript to e.g. Kang SW et al., 2007 (PMID17851169) showing this exactly for rabbit-derived chondrocytes….

Thank you for this valuable comment. The authors reviewed the excellent work of Kang SW et al. and gladly include their work on chondrocyte dedifferentiation in a rabbit model in our manuscript and added the according reference. The revised passage reads as follows (line 273-274): 

Higher passage is known to have highly deteriorating effects on chondrocyte phenotypes. Kang et al. already described passage 5 to exhibit a strong expression of type I collagen (ref. [5] Kang et al.)

Was an autologous approach used in the study? It is not clearly enough stated in the manuscript. Otherwise it is recommended to write „allogenic“ or indeed „autologous rabbit model“ e.g. in the title. It is also not completely clear whether indeed a chondral defect model was used and not an osteochondral one. It should be stated in the abstract.

Thank you for this important note. In our study, an autologous chondrocyte transplantation approach was applied. We changed the title and the corresponding passages to point this out (line 2-4, 32 and 377):

Chondrocyte culture parameters for matrix-assisted autologous chondrocyte implantation affect catabolism and inflammation in a rabbit model.

After three different cell passages (P1, P3, P5), cells were seeded on 3D collagen matrices at three different densities (2x105/matrix, 1x106/matrix, 3x106/matrix) combined with two different membrane-holding-times (5 hours, 2 weeks) prior autologous transplantation.

For the autologous CSMs in vivo implantation, a medial para-patellar arthrotomy was performed on the left knee.

It is not always detailed whether the mentioned differences between data are indeed significant or not.

The authors agree to this notion and added the following passage to our manuscript (line 454-455):

If our results yielded statistical significance, we explicitly refer to this in the results section of this manuscript. If not mentioned explicitly, the results are not statistically significant. 

The terms „implantation/implanted“ and „transplant/ation“ (e.g. line 190 and line 192) seems to be mixed throughout the manuscript, please provide a rationale.

Thank you for this correct remark. The term “transplantation” was used when talking about the whole assessment with chondrocyte extraction and implantation after preparation. We used “implantation” to focus only on the act of implanting the cell-seeded matrix into the prepared defects.

Abstract

„contribute“ write „to“ instead of „in“

We thank the reviewer for this correction. According to the reviewer’s remark, we changed the phrase “contribute in” to “contribute to” (line 24).

Line 30: the dimensions (thickness, diameter: volume) of the collagen membrane should be stated to get a relation to the cell numbers listed here

As requested by the esteemed reviewer we provided the approx. volume of the matrix within the abstract, line 30:

“(approx. 27 mm3)“

As technically described within the methods section, the defect size was 4 mm in diameter and 2 mm in depth and was applied with a customized drill to create standardized chondral defects within the trochlear groove. The dimensions of the dry membrane were shaped according to the stated defect size and correspond to 25,133 mm3 of volume. The membrane used is known to increase its dimensions up to 10% when moistened. Therefore, we approximated the volume to 27 mm3.

Line 32: „Two defects“, please write already here whether they are „critically sized (dimension!)“ or not. Compare „4 mm“ line 345, how about the depth?

Line 33: Write also already here, where the defects are exactly localized (trochlea, med./lat. condyles?)

We appreciate these two requests and extended line 33/34:

“Two defects/knee/animal were performed”

we changed into

Two defects/knee/animal were created in the trochlear groove (defect dimension: Æ4 mm x 2 mm)”

There is strong debate in the literature if a 4mm rabbit defect can be regarded as critical or not. From our intraoperative experience, we do consider 4mm a critical size defect. Concerning defect depth, care was taken not to create osteochondral lesions.

Introduction

TNFα, IL-1β, MMP1-3: please explain why exactly these targets were investigated in the study. Please consider also the important role of proinflammatory cytokines in the early healing process and that of MMPs in the later remodeling process. Proinflammatory cytokines are not exclusively „bad guys“

The authors really appreciate this excellent comment by reviewer 1, for the here stated issue is very important while often being discussed and misinterpreted. Like reviewer 1, we do not consider proinflammatory cytokines to be “exclusively bad guys”, but to play an immensely important role within the complex cascades of inflammation. As the superfamily of zinc-dependent proteinases does take part in degrading as well as cell-redeeming processes, proinflammatory cytokines interact with them and with each other, thus directly as well as indirectly providing their share to tissue homeostasis, degrading, remodelling as well as repair i.e. regeneration.

According to reviewer 1’s important comment, we rephrased and extended the introduction (line 76-79):

As the superfamily of zinc-dependent proteinases might take part in degrading as well as cell redeeming processes, proinflammatory cytokines interact with them and with each other, thus directly as well as indirectly provide their share to tissue homeostasis, degrading, remodelling as well as repair i.e. regeneration.

Fig. 3:

Y axis: if correct, write gene expression.

Following the honoured reviewer’s suggestion, we extended the designation of the Y axis with the term “expression”, amending the figure description “mRNA expression” (line 115 and 152)

Fig. 7: membrane holding time n=2 – is it sufficient for statistics?

We thank the reviewer for this important question. For our study, we assumed that membrane holding time amongst others affects the expression of diverse cytokines and proteinases. To evaluate this presumption, we compared two different membrane holding times. The number of groups was limited to two in order to prevent from multiple regression. Furthermore: By multiplication with all the other diverse groups this animal study reached a highly strong statistical power. Also, the selected membrane holding times are very closely connected to those currently practiced during clinical m-ACI.

Discussion

Comment on the fact that gene expression is investigated in regard to MMPs but not activated protein levels.

The authors appreciate this interesting request. The here presented study explicitly investigated mRNA expression patterns of MMPs and cytokines and their potential dependency of in vitro cell culture settings. mRNA levels do give highly specific and very sensitive detail answers. Against that, true protein level sometimes requires months to be truly and correctly interpreted. This is out of scope of this study.

Within further studies, it would be very interesting to further investigate if the here presented results are true for the activated protein level fraction of MMPs as well.

We commented on this fact and extended the discussion on it (line 191-197)

The here-presented study explicitly investigated mRNA expression patterns of MMPs and cytokines and their potential dependency of in vitro cell culture settings. Within further studies, it would be very interesting to further investigate if the here presented results are true for the activated protein level fraction of MMPs as well.

First paragraph: The different regulation of TNFα in contrast to IL-1β is highly interesting and should be discussed in more detail (develop a hypothesis).

„membrane holding time “why is this parameter so important: provide a hypothesis.

How about the effect of particular the biomaterial used in this study and the previous study from 2011? It should be discussed why the authors could generalize their results to other biomaterials? ….

1.     We would like to thank the reviewer for the important input and kindly refer to line 181-185, where we focused on the contrary courses of TNFα and IL-1 β.

2.     As also already discussed by Salzmann et al. (ref. [3] Salzmann et al.), the membrane holding time represents one of severe items to create optimal expression patterns regarding chondrogenic differentiation and thus improve the patient’s outcome in ACI.

3.     Thank you for this important question. In our studies, we used a biomaterial that is broadly applied and acknowledged. Of course, there are other biomaterials that use other approaches of basic material, structure and density. We would not generalize the here used biomaterial, but we think that it could be a reference to generalize the use of a similar basis of biomaterials.

Between the first and second paragraph of the discussion there seems to be a break – please create a bridge.

We extended the first paragraph of the discussion in order to bridge the break remarked by the honoured reviewer and to preserve the arc of suspense provided by the topic. (line 194-197)

The latter notion combined with the results presented here could provide further insights to the cascades of early healing response and/or tissue degradation following cartilage damage and potentially deterioration of joint homeostasis.

Line 230: The sentence requires a citation for support.

We sincerely thank the reviewer for the disclosure of this unfortunate mistake.

We corrected the passage for proper statement and reference (line 255-256):

Today it is known that a short membrane-holding time has beneficial effects on graft maturation and differentiation (ref. [3] Salzmann et al.).

Line 236: „same osteochondral rabbit animal model“. Be aware that elsewhere in the manuscript the used model is designated as „chondral“….please correct!

We thank the reviewer for making us aware of this blunder and we adapted the named term (line 262) accordingly:

In a recent study using an osteochondral rabbit animal model, no positive effect on cartilage repair was shown in vitro and in vivo between different cell seeding densities.

Materials and methods

4.1.: „femal post puberty“. Please mention the approximate age of the animals and whether epiphyses are closed or not as a parameter to estimate roughly musculoskeletal maturity. Compare line 317: 3.5 kg BW.

According to this remark, we supplemented the concerned phrase with details about the included animals (line 348-349). The rabbits were obtained from Charles River (Kisslegg, Germany) who ensured that growth plates were closed in each animal. Of course, we verified this circumstance by assessing the Stroh sign (ref. 2):

All rabbits were postpubescent with closed growth plates which was verified by applying the Stroh sign (ref. [2] Bujalska et al).

Line 275: provide exact dimension of the membrane it required to understand the cell numbers, compare line 337

We appreciate this advice and added the volume of the membrane to our revised manuscript (line 305):

Subsequently, chondrocytes were seeded on the 3D collagen membrane (Chondro-Gide®, Geistlich, Wollhusen, Switzerland) with a volume of approximately 25 mm3 at different cell densities (i.e. 2x105/matrix = C1, 1x106/matrix = C2 or 3x106/matrix = C3).

Why was no autologous rabbit serum used for culturing? Compare line 334: fetal bovine serum.

Thank you for this comment. We preferred fetal bovine serum for our study as it has a rather long history of usage with reliable standards in fabrication (ref. [1] Shah et al.) whereas rabbit serum is applied infrequently and consequently expertise is lacking. Moreover, a significant influence of the used serum on the outcome did not appear probable to us.

Line 323: „full thickness cartilage pieces“. How much cartilage was explanted for cell isolation? Was it only removed from a non-weight bearing area of the trochlea (e.g. uppermost cranial part?). Or was the whole joint cartilage removed? How about the welfare of the animals after removing completely the trochlear joint cartilage?

The authors appreciate this interesting annotation.

For cell isolation, we removed cartilage from the non-weight bearing area of the lateral trochlea. This is a typical procedure in order to harvest cartilage in rabbits (and also a matter of debate), but the described procedure is widely performed and acknowledged in humans as well. A specification of the exact size of the extracted cartilage is not possible and in our view, not essential. Care was taken, that all animals reached full weight-bearing within one day post-surgery.

We adapted the specific section in the revised manuscript to clarify the conditions of the cartilage explantation (line 354-358):

Hereafter, full-thickness cartilage pieces were harvested from the non-weight-bearing part of the trochlea femoris and the joint was then closed in layers. Carprofen (4 mg/kg once a day) and buprenorphine (0.03 mg/kg twice a day) were subcutaneously administered as post-operative analgesics for 3 days and the animals were then further kept under standard housing conditions. All animals reached full weight-bearing within one day postoperative. 

Line 341: membrane-holding time. Was the seeding efficacy determined. How many cells survived and adhered on the scaffolds?

No exact cell count was performed as it did not appear relevant to us. Yet, from earlier and other experiments from our group and also comparing to clinical data cell survival within membranes is commonly around 90%.

Line 345: add „diameter“and describe the depth of the defect.

Also at this point, we complemented the delineation of the defect (line 378):

Two chondral defects (4 mm in diameter, 2 mm in depth) were created in the left knee of each animal within the center of the trochlea groove.

Line 345: „chondral defects” I am not sure whether the effects are indeed chondral?

We thank the reviewer for this important question. The defects that were created in the rabbit’s knee were actually chondral defects as is apparent e.g. from line 377.

Line 347: „press fitting”– I think it is not possible to achieve in a chondral defect in the rabbit model sufficient fixation by press fitting and also not by fibrin sealant…it would be possible in an osteochondral defect in the rabbit.

Thank you for this critical remark. As you mentioned, we neither expected to achieve sufficient fixation by press fitting or fibrin adhering alone. In the end, we experienced that a combination of both techniques provides certain fixation which was also checked intraoperatively by moving the joint. We added the conducted test to the according section (line 381-385).

The defects were then sealed with fibrin glue (Baxter, Vienna, Austria) whereby a certain fixation of the membrane could be obtained. Next, the patella was relocated and the joint was bended and stretched in full range of motion for ten times. The patella was dislocated again to revise the fixation. Finally, the joint was closed in layers.

Please show an OP image of the defect with the scaffold.

For the honoured reviewer's request, we here provide an original high resolution intra-operative picture of our press-fit implanted membrane according to the dimensions described earlier.

Line 410:

Figure 9: Intraoperative view on the chondral defect filled with the cell-seeded matrix.

Line 349: please discuss the effect of the fibrin because it is probably human-derived.

We thank the reviewer for this note and stated the effect of the fibrin glue that was applied to seal the defects (line 381-383):

The defects were then sealed with fibrin glue (Baxter, Vienna, Austria), whereby a certain fixation of the membrane could be obtained.

References:

1.    Shah G: Why do we still use serum in the production of biopharmaceuticals? Dev Biol Stand 1999, 99:17-22.

2.    Bujalska G: Studies on the European hare. VI. Comparison of different criteria of age. Acta theorilogica 10:1-9 · August 1965 DOI: 10.4098/AT.arch.65-1

3.    Salzmann, G.M.; Sauerschnig, M.; Berninger, M.T.; Kaltenhauser, T.; Schonfelder, M.; Vogt, S.; Wexel, G.; Tischer, T.; Sudkamp, N.; Niemeyer, P., et al. The dependence of autologous chondrocyte transplantation on varying cellular passage, yield and culture duration. Biomaterials 2011, 32, 5810-5818, doi:S0142-9612(11)00495-9 [pii] 10.1016/j.biomaterials.2011.04.073.

4.       Boehme KA, Rolauffs B. Onset and Progression of Human Osteoarthritis-Can Growth Factors, Inflammatory Cytokines, or Differential miRNA Expression Concomitantly Induce Proliferation, ECM Degradation, and Inflammation in Articular Cartilage? Int J Mol Sci. 2018 Aug 3;19(8). pii: E2282. doi: 10.3390/ijms19082282. PMID: 30081513

5: Kang SW, Yoo SP, Kim BS. Effect of chondrocyte passage number on histological aspects of tissue-engineered cartilage. Biomed Mater Eng. 2007;17(5):269-76.

Reviewer 2 Report

The article Ms. Ref. No.: ijms-456746 entitled “Chondrocyte culture parameters for matrix-assisted chondrocyte implantation affect catabolism and inflammation in a rabbit model" is interesting. However, the paper can be suitable for publication in the “International Journal of Molecular Sciences after minor revisions.

The manuscript is well written and well detailed. However, some points need to be ameliorated.

- The description of osteoarthritis pathogenesis can be improved and associated with more updated references.

-Please include in the figure legends the details about the statistical analysis for “*” and “°”, it’s not clear.

Author Response

Point-by-point response

Reviewer#2

The description of osteoarthritis pathogenesis can be improved and associated with more updated references.

We thank the reviewer for this note and updated references throughout the whole manuscript accordingly. We also extended the introduction (line 66-75):

Most recently, Karen Boehme and colleagues presented a review of the literature focusing on onset and progression of human OA – with the highly appreciated very focus on the often discussed growth factors as well as inflammatory cytokines (ref. [4] Boehme et al.). The named review disclosed only fibroblast growth factor 2 (FGF2) to be capable of inducing all major key events of early OA, i.e. the proliferation of cartilage-resident cells, the degradation of extracellular matrix components and the inflammation so far throughout known literature. This conclusion happened to be a major trigger in order to further investigate the basics of the pathophysiology of OA within the realms of the study presented here.

Halbwirth et al. reported that chondrocytes themselves play a major role in the pathophysiology of OA.

Please include in the figure legends the details about the statistical analysis for “*” and “°”, it’s not clear.

Thank you for this remark. Outliers from the data are marked as * and ° in the presented figures. As a normal distribution is not given among data, the data presentation was carried out using boxplots where outliers where of course not considered when generating the median. To follow your advice, we supplemented the figure legends (line 117, 121, 153 and 164).

Round 2

Reviewer 1 Report

The conclusions of the study is only based on the profile of selected cytokines and MMPs without a feedback concerning healing based on the macroscopical healing outcome, clinical, histological or even protein data. Hence, it remains superficial with limited predication. The authors explained that they cannot change any more the original (and now locked) study protocol for ethical reasons, This study was not planed comprehensive enough for my opinion especially in view of the large animal number consumed in the study. This still weakens this study substantially.

Nevertheless, the authors mention this limitation now in the methods section (Line 297).

Controls have been discussed now. The limitations of the study are inserted in the discussion now.

The authors discussed the comments of the reviewer and changed the manuscript accordingly. The manuscript benefs from the changes performed.

Some remaining issues:

Line 30: defect volume approx. 27 mm2 vs: volume of the collagen membrane: Collagen membrane (25 mm, Line 306). Please explain the difference! The filled defect should reach the Level of the surrounding cartilage.

I am still wondering that the defect model should be a chondral one. Ahern et al., (2009) reported a cartilage thickness of 0.3 mm in the mature rabbit model (medial femoral condyle). The trochlear cartilage might not be substantial thicker also for my own experience. Therefore, 2 mm depth (line 378) might represent rather an osteochondral defect than a chondral defect (Line 389 and 410). In fig. 9: also bleeding around the implant is visible suggesting access to the bone marrow.

This should be considered. Please find a reference/evidence for the cartilage thickness of 2 mm in the rabbits used in this study.

Author Response

R2: Point-by-point response

Reviewer#1

Line 30: defect volume approx. 27 mm2 vs: volume of the collagen membrane: Collagen membrane (25 mm, Line 306). Please explain the difference! The filled defect should reach the Level of the surrounding cartilage.

We thank the reviewer for this comment. The difference between the two named volumes of the collagen membrane results from the moistening that is known to increase the membrane’s volume by up to 10 %. The volume of 25 mm3 is an approximate value of the dry membrane.

We changed the description of the membrane in line 30 to “approx. 25 mm3 to remove the ambiguity that was kindly pointed out by the reviewer.

I am still wondering that the defect model should be a chondral one. Ahern et al., (2009) reported a cartilage thickness of 0.3 mm in the mature rabbit model (medial femoral condyle). The trochlear cartilage might not be substantial thicker also for my own experience. Therefore, 2 mm depth (line 378) might represent rather an osteochondral defect than a chondral defect (Line 389 and 410). In fig. 9: also bleeding around the implant is visible suggesting access to the bone marrow.

This should be considered. Please find a reference/evidence for the cartilage thickness of 2 mm in the rabbits used in this study.

Thank you for this important note. As you already stated, no exact description of the trochlear cartilage and its depth in the rabbit model can be found in the literature – at least to the best of our knowledge. In several publications, the depth of osteochondral defects (3 to 5 mm) created in the trochlear groove of mature rabbits is described. The depths created there by far exceed the realms of the defects created in our study setup [1-6].

In our study, accurate care was taken not to harm the subchondral layer while creating the defects. As the collagen membranes with a ductile thickness of approximately 2 mm precisely filled the chondral defects, we assumed that the depth of the defects amounted to a maximum of 2 mm, respectively.

Consequently, the minimal bleeding that can be seen in figure 9 certainly did not origin from the subchondral bone, but from the cutaneous/subcutaneous tissue and therefore has to be considered as plain blood recess that usually occurs during surgical procedures.

References:

1.         Oshima T, Nakase J, Toratani T, Numata H, Takata Y, Nakayama K, Tsuchiya H: A Scaffold-Free Allogeneic Construct From Adipose-Derived Stem Cells Regenerates an Osteochondral Defect in a Rabbit Model. Arthroscopy 2019, 35(2):583-593.

2.         Dwivedi G, Chevrier A, Alameh MG, Hoemann CD, Buschmann MD: Quality of Cartilage Repair from Marrow Stimulation Correlates with Cell Number, Clonogenic, Chondrogenic, and Matrix Production Potential of Underlying Bone Marrow Stromal Cells in a Rabbit Model. Cartilage 2018:1947603518812555.

3.         Park IS, Jin RL, Oh HJ, Truong MD, Choi BH, Park SH, Park DY, Min BH: Sizable Scaffold-Free Tissue-Engineered Articular Cartilage Construct for Cartilage Defect Repair. Artif Organs 2019, 43(3):278-287.

4.         Chang NJ, Erdenekhuyag Y, Chou PH, Chu CJ, Lin CC, Shie MY: Therapeutic Effects of the Addition of Platelet-Rich Plasma to Bioimplants and Early Rehabilitation Exercise on Articular Cartilage Repair. Am J Sports Med 2018, 46(9):2232-2241.

5.         Sawada Y, Sugimoto A, Osaki T, Okamoto Y: Ajuga decumbens stimulates mesenchymal stem cell differentiation and regenerates cartilage in a rabbit osteoarthritis model. Exp Ther Med 2018, 15(5):4080-4088.

6.         Zylinska B, Stodolak-Zych E, Sobczynska-Rak A, Szponder T, Silmanowicz P, Lancut M, Jarosz L, Rozanski P, Polkowska I: Osteochondral Repair Using Porous Three-dimensional Nanocomposite Scaffolds in a Rabbit Model. In Vivo 2017, 31(5):895-903.